# Impact of periprocedural morphine use on mortality in STEMI patients treated with primary PCI

Dominika Domokos[1], Andras Szabo[2], Gyongyver Banhegyi[3], Laszlo Major[4], Robert Gabor Kiss[4], David Becker[1], Istvan Ferenc Edes[1], Zoltan Ruzsa[1,5], Bela Merkely[1], Istvan Hizoh[1]*

1 Heart and Vascular Center, Semmelweis University, Budapest, Hungary, 2 Department of Anesthesiology and Intensive Care, Semmelweis University, Budapest, Hungary, 3 Independent Researcher, Budapest, Hungary, 4 Department of Cardiology, Medical Center, Hungarian Defense Forces, Budapest, Hungary, 5 Department of Invasive Cardiology, Bacs-Kiskun County University Teaching Hospital, Kecskemet, Hungary

* istvan.hizoh@alumni.uni-heidelberg.de

**Data Availability Statement:** All relevant data are within the paper and its Supporting Information files.

## Abstract

### Background

Intravenous morphine (MO) decreases the effect of all oral platelet $P2Y_{12}$ receptor inhibitors in vitro and observational reports suggest that its use may be associated with larger infarct size. Yet, there are limited data available about the impact of this interaction on clinical outcomes. We studied the effect of MO on mortality in ST-segment elevation myocardial infarction (STEMI) patients treated with primary PCI using a prospective registry.

### Methods

Of the 1255 patients who underwent primary PCI, 397 received MO based on physician's judgment. Clopidogrel was used as $P2Y_{12}$ receptor antagonist in all cases. Median follow-up time was 7.5 years with 457 deaths. To adjust for confounding, two propensity score-based procedures were performed: 1 to 1 matching (PSM, 728 cases), and inverse probability of treatment weighting (IPTW) retaining data from all patients. Primary outcome measure was time to all-cause death, whereas predischarge left ventricular ejection fraction (LVEF) was used as secondary end point.

### Results

An adequate balance on baseline covariates was achieved by both methods. We found no difference in survival as the HR (MO/no MO) was 0.98 (95% confidence interval [CI]: 0.76–1.26), p = 0.86 using PSM and 1.01 (95% CI: 0.84–1.23), p = 0.88 with IPTW. Likewise, distributions of LVEFs were similar using either methods: with PSM, median LVEFs were 50.0% (interquartile range [IQR]: 43.0%–55.3%) vs 50.0% (IQR: 42.0%–55.0%) in the no MO and MO groups, respectively (p = 0.76), whereas using IPTW, they were 50.0% (IQR: 42.5%–55.0%) vs 50.0% (IQR: 41.0%–55.0%), respectively (p = 0.86).

**Funding:** The present work was supported by the Hungarian Artificial Intelligence National Laboratory (NKFIH-870-1/2020, a grant from the National Research, Development and Innovation Office of Hungary to BM). The funder had no role in study design, data collection and analysis, decision to publish, or preparation of the manuscript.

**Competing interests:** The authors have declared that no competing interests exist.

## Conclusions

Our data suggest that morphine use may have no impact on long-term mortality and on pre-discharge ejection fraction in STEMI patients treated with primary PCI.

## Introduction

In the setting of ST-segment elevation myocardial infarction (STEMI), intravenous (IV) morphine (MO) is traditionally employed to relieve pain, reduce pulmonary congestion, and anxiety. Though the efficacy and safety of morphine use were not studied in randomized clinical trials, both European and American guidelines on STEMI recommend its application in these conditions based on expert consensus [1,2]. Nevertheless, according to recent studies, morphine delays and decreases the effects of all currently available oral platelet $P2Y_{12}$ receptor inhibitors (i.e., clopidogrel, prasugrel, and ticagrelor) in vitro [3–9] which may result in poorer myocardial reperfusion [10] and larger infarct size [11]. In the light of that, the current European guidelines add a note of caution that the diminished effects of clopidogrel, ticagrelor, and prasugrel may lead to early treatment failure [1]. Yet, there are few data available about the impact of this interaction on clinical outcomes and the effect on long-term mortality is barely investigated [12–19]. Therefore, we studied the impact of periprocedural morphine application on all-cause mortality in STEMI patients treated with primary percutaneous coronary intervention (PCI) using a prospective registry.

## Materials and methods

### Study design, outcome measures

We analyzed observational data of 1255 consecutive STEMI patients of a single-center prospective registry who were treated with primary PCI from September 2007 through December 2011. Of them, 397 (31.6%) received morphine intravenously based on physician's judgment in the periprocedural period. The decision to use morphine during primary PCI was independent of the present research. To control for biased baseline covariates, two distinct propensity score-based methods were performed: 1 to 1 nearest neighbor propensity score matching (PSM) yielding a total of 728 patients and inverse probability of treatment weighting (IPTW) retaining data from all patients (for details see Statistical Analysis and Results sections). Primary outcome measure of the study was time to all-cause death, whereas predischarge left ventricular ejection fraction (LVEF) assessed by echocardiography was used as secondary end point. All patients were followed-up by means of hospital records, follow-up visits, telephone interviews, and records of the National Health Insurance Fund. No patients were lost to follow-up. Median follow-up time was 7.5 years. All the 1255 cases were complete cases with no missing data. Patient data were prospectively collected in the Medical Center, Hungarian Defense Forces, Budapest, Hungary according to applicable laws and regulations: 1997 XLVII Act on the Handling and Protection of Health and Related Personal Data which was modified by Acts CCXLIV of 2013 and CXVIII of 2018 and Decrees 15/2014 and 49/2018 of the Ministry of Human Resources of Hungary. A formal approval of data collection by the institutional review board was not required because all Hungarian health care providers are obliged by the above mentioned laws to provide anonymized data of all patients with myocardial infarction for the prospective National Registry of Myocardial Infarction. The use of these institutional anonymized data for this specific scientific research was approved by the head of the

institution. All patients gave written informed consent to be available for follow-up and the research was conducted according to the principles of the Declaration of Helsinki.

## Procedure

Application of intravenous morphine in the periprocedural period (i.e., from onset of the symptoms to two hours following the PCI) was left to the physician's discretion and was independent of the present analysis. During the study, morphine hydrochloride (molecular weight: 321.8 g/mol) was used exclusively, morphine sulfate (molecular weight: 668.8 g/mol) was not applied. Primary PCI was performed using standard techniques. The arterial sheath was removed immediately after the procedure. Bleeding from the radial artery was stopped using the TR Band (Terumo Europe, Leuven, Belgium), while the femoral artery was closed by the FemoSeal device (St. Jude Medical, St. Paul, MN). In cases of persistent femoral artery bleeding, manual compression was applied. All patients were treated with acetylsalicylic acid and a loading dose of 600 mg clopidogrel and discharged on dual antiplatelet therapy for at least 12 months. Successful PCI was defined as <50% diameter stenosis with a final TIMI flow grade≥2. Interventional cardiologists were high-volume operators (i.e., >200 PCIs/year) skilled in both transfemoral and transradial techniques. Left ventricular ejection fraction was assessed by echocardiography within 48 hours after the index procedure.

## Statistical analysis

For descriptive statistics, variables in 2×2 contingency tables were assessed using Fisher's exact test. Categorical data in 2×k tables were analyzed using the unordered chi-squared test or, to detect linear trend, the chi-squared test for trend. As none of the continuous variables showed normal distribution, the Wilcoxon rank sum test was applied for their comparisons. A two-tailed p value less than 0.05 was considered statistically significant.

To adjust for confounders, two distinct propensity score-based techniques were applied [20]. We used 1 to 1 nearest neighbor propensity score matching with a caliper width of 0.2 to estimate the average treatment effect for the treated (ATT) yielding a total of 728 cases [21]. In addition, we also assessed the average treatment effect (ATE) by inverse probability of treatment weighting (IPTW) using stabilized weights retaining data from all patients [20,22]. The propensity score model included all measured baseline covariates listed in Table 1 that could affect treatment assignment and/or are known to be associated with the primary end point. Balance on baseline covariates between the treated and control groups was evaluated using absolute standardized differences [23]. A value less than 0.1 was considered as an acceptable standardized bias. Absolute risk differences in all-cause mortality were captured by Kaplan-Meier survival curves which were compared using log-rank tests. The relative change in the hazard of death was estimated using univariable Cox models as suggested by Austin [20,24]. As to the secondary outcome measure, distributions of predischarge LVEFs in the treated and control groups were compared by rank tests. All statistical analyses and graphical interpretation of the results were carried out with R version 4.0.2 (R Foundation for Statistical Computing, Vienna, Austria). For further details, please see S1 Appendix.

## Results

### Patient characteristics, propensity score model, morphine dose

Baseline demographic, clinical, and procedural characteristics of treated and control patients in the original, matched, and weighted samples are summarized in Table 1. Systematic differences between treated and untreated patients in the original cohort have been eliminated in

**Table 1. Baseline demographic, clinical, and procedural characteristics.**

| Variable | Original Sample | | | | Matched Sample | | | Weighted Sample | | |
|---|---|---|---|---|---|---|---|---|---|---|
| | No Morphine (n = 858) | Morphine (n = 397) | Absolute Standardized Difference | p Value | No Morphine (n = 364) | Morphine (n = 364) | Absolute Standardized Difference | No Morphine (n = 860) | Morphine (n = 394) | Absolute Standardized Difference |
| Age Median (IQR) (years) | 63.0 (54.0–73.0) | 62.0 (54.0–72.0) | 0.0499 | 0.38 | 63.0 (54.0–73.0) | 62.0 (54.0–72.0) | 0.0177 | 63.0 (54.0–72.0) | 62.7 (54.0–72.0) | 0.0010 |
| BMI Median (IQR) (kg/m$^2$) | 27.2 (24.2–30.4) | 27.0 (24.3–30.5) | 0.0286 | 0.86 | 27.0 (24.4–30.4) | 26.8 (24.2–30.5) | 0.0664 | 27.0 (24.2–30.3) | 26.7 (24.2–30.1) | 0.0524 |
| Female | 303 (35.3%) | 141 (35.5%) | 0.0042 | 0.95 | 132 (36.3%) | 129 (35.4%) | 0.0172 | 304 (35.3%) | 146 (37.1%) | 0.0361 |
| Hypertension | 594 (69.2%) | 275 (69.3%) | 0.0008 | 1.00 | 253 (69.5%) | 250 (68.7%) | 0.0178 | 596 (69.3%) | 271 (68.8%) | 0.0097 |
| Diabetes mellitus | 219 (25.5%) | 90 (22.7%) | 0.0667 | 0.29 | 90 (24.7%) | 84 (23.1%) | 0.0385 | 217 (25.2%) | 102 (25.8%) | 0.0124 |
| Verified dyslipidemia | 345 (40.2%) | 154 (38.8%) | 0.0290 | 0.66 | 140 (38.5%) | 141 (38.7%) | 0.0056 | 342 (39.7%) | 151 (38.2%) | 0.0306 |
| Current smokers | 306 (35.7%) | 178 (44.8%) | 0.1877 | 0.0022 | 147 (40.4%) | 162 (44.3%) | 0.0843 | 335 (38.9%) | 159 (40.4%) | 0.0305 |
| Peripheral artery disease | 63 (7.3%) | 26 (6.5%) | 0.0312 | 0.64 | 24 (6.6%) | 25 (6.9%) | 0.0108 | 63 (7.3%) | 33 (8.4%) | 0.0414 |
| Cerebrovascular disease | 72 (8.4%) | 27 (6.8%) | 0.0600 | 0.37 | 24 (6.6%) | 26 (7.1%) | 0.0207 | 66 (7.7%) | 27 (6.9%) | 0.0295 |
| Congestive heart failure | 46 (5.4%) | 10 (2.5%) | 0.1464 | 0.03 | 12 (3.3%) | 10 (2.7%) | 0.0283 | 38 (4.4%) | 13 (3.2%) | 0.0621 |
| Previous myocardial infarction | 106 (12.4%) | 42 (10.6%) | 0.0557 | 0.40 | 30 (8.2%) | 41 (11.3%) | 0.0948 | 101 (11.8%) | 47 (11.9%) | 0.0025 |
| Previous percutaneous coronary intervention | 63 (7.3%) | 26 (6.5%) | 0.0312 | 0.64 | 19 (5.2%) | 25 (6.9%) | 0.0648 | 63 (7.3%) | 25 (6.3%) | 0.0416 |
| Previous coronary artery bypass graft surgery | 16 (1.9%) | 10 (2.5%) | 0.0446 | 0.52 | 7 (1.9%) | 9 (2.5%) | 0.0375 | 18 (2.1%) | 8 (2.0%) | 0.0072 |
| Chronic renal failure | 33 (3.8%) | 10 (2.5%) | 0.0756 | 0.25 | 11 (3.0%) | 10 (2.7%) | 0.0156 | 29 (3.4%) | 10 (2.7%) | 0.0429 |
| Baseline Creatinine Median (IQR) (micromol/L) | 79.0 (67.0–98.0) | 79.0 (65.0–94.0) | 0.0256 | 0.50 | 78.0 (66.0–96.0) | 79.0 (65.0–93.3) | 0.0219 | 78.0 (67.0–97.0) | 50.0 (41.0–55.0) | 0.0438 |
| Chronic obstructive pulmonary disease | 64 (7.5%) | 28 (7.1%) | 0.0156 | 0.91 | 21 (5.8%) | 27 (7.4%) | 0.0635 | 64 (7.4%) | 26 (6.5%) | 0.0333 |
| Prehospital heparin | 481 (56.1%) | 296 (74.6%) | 0.3958 | <0.0001 | 260 (71.4%) | 266 (73.1%) | 0.0353 | 534 (62.0%) | 248 (63.0%) | 0.0198 |
| Prehospital clopidogrel | 577 (67.2%) | 331 (83.4%) | 0.3804 | <0.0001 | 297 (81.6%) | 298 (81.9%) | 0.0065 | 624 (72.5%) | 291 (73.8%) | 0.0298 |
| Onset-to-door time Median (IQR) (hours) | 3.6 (2.0–6.0) | 2.5 (2.0–4.0) | 0.3993 | <0.0001 | 3.0 (2.0–4.5) | 2.68 (2.0–4.0) | 0.0254 | 3.0 (2.0–5.5) | 3.5 (2.0–5.0) | 0.0093 |
| Door-to-balloon time Median (IQR) (min.) | 47.0 (32.0–75.0) | 45.0 (30.0–68.0) | 0.1739 | 0.0069 | 45.0 (30.0–68.0) | 45.0 (29.8–67.0) | 0.0242 | 46.0 (30.0–71.0) | 49.6 (30.0-70-0) | 0.0481 |
| ECG localization | | | | 0.20 | | | | | | |
| anterior | 347 (40.4%) | 181 (45.6%) | 0.1040 | | 160 (44.0%) | 164 (45.1%) | 0.0222 | 362 (42.1%) | 159 (40.3%) | 0.0367 |

*(Continued)*

**Table 1.** (Continued)

| Variable | Original Sample | | | | Matched Sample | | | Weighted Sample | | |
| --- | --- | --- | --- | --- | --- | --- | --- | --- | --- | --- |
| | No Morphine (n = 858) | Morphine (n = 397) | Absolute Standardized Difference | p Value | No Morphine (n = 364) | Morphine (n = 364) | Absolute Standardized Difference | No Morphine (n = 860) | Morphine (n = 394) | Absolute Standardized Difference |
| inferior | 455 (53.0%) | 195 (49.1%) | 0.0782 | | 186 (51.1%) | 183 (50.3%) | 0.0165 | 445 (51.7%) | 215 (54.5%) | 0.0558 |
| posterior/lateral | 56 (6.5%) | 21 (5.3%) | 0.0524 | | 18 (4.9%) | 17 (4.7%) | 0.0116 | 54 (6.2%) | 21 (5.2%) | 0.0412 |
| Cardiac arrest on or prior to admission | 81 (9.4%) | 22 (5.5%) | 0.1484 | 0.02 | 23 (6.3%) | 21 (5.8%) | 0.0209 | 70 (8.1%) | 33 (8.3%) | 0.0066 |
| Heart rate Median (IQR) (1/min) | 80.0 (69.0–90.0) | 78.0 (67.0–90.0) | 0.0407 | 0.55 | 80.0 (68.0–90.0) | 79.0 (67.0–90.0) | 0.0108 | 80.0 (69.0–90.0) | 78.0 (67.0–90.0) | 0.0119 |
| Systolic blood pressure Median (IQR) (mmHg) | 130.0 (110.0–148.0) | 130.0 (110.0–145.0) | 0.0260 | 0.66 | 130.0 (111.5–141.2) | 130.0 (110.0–140.5) | 0.0486 | 130.0 (110.0–145.6) | 130.0 (110.0–145.0) | 0.0045 |
| Killip class | | | | 0.95 | | | | | | |
| 1 | 721 (84.0%) | 317 (79.8%) | 0.1088 | | 300 (82.4%) | 294 (80.8%) | 0.0429 | 708 (82.3%) | 330 (83.8%) | 0.0381 |
| 2 | 54 (6.3%) | 48 (12.1%) | 0.2015 | | 34 (9.3%) | 39 (10.7%) | 0.0477 | 70 (8.1%) | 32 (8.2%) | 0.0021 |
| 3 | 13 (1.5%) | 11 (2.8%) | 0.0867 | | 12 (3.3%) | 11 (3.0%) | 0.0190 | 21 (2.4%) | 9 (2.2%) | 0.0147 |
| 4 | 70 (8.2%) | 21 (5.3%) | 0.1146 | | 18 (4.9%) | 20 (5.5%) | 0.0220 | 61 (7.1%) | 23 (5.8%) | 0.0525 |
| Intra-aortic balloon pump | 52 (6.1%) | 17 (4.3%) | 0.0803 | 0.23 | 16 (4.4%) | 16 (4.4%) | 0.0000 | 46 (5.4%) | 17 (4.4%) | 0.0444 |
| Mechanical ventilation | 120 (14.0%) | 32 (8.1%) | 0.1899 | 0.0028 | 32 (8.8%) | 32 (8.8%) | 0.0000 | 107 (12.4%) | 48 (12.1%) | 0.0093 |
| Glycoprotein IIb/IIIa receptor inhibitor | 686 (80.0%) | 351 (88.4%) | 0.2332 | 0.0002 | 313 (86.0%) | 320 (87.9%) | 0.0530 | 709 (82.4%) | 320 (81.2%) | 0.0331 |
| Transradial primary percutaneous coronary intervention | 742 (86.5%) | 363 (91.4%) | 0.1585 | 0.01 | 334 (91.8%) | 332 (91.2%) | 0.0176 | 756 (87.9%) | 342 (86.8%) | 0.0361 |
| Access site conversion | 31 (3.6%) | 17 (4.3%) | 0.0343 | 0.64 | 16 (4.4%) | 15 (4.1%) | 0.0141 | 34 (4.0%) | 15 (3.7%) | 0.0158 |
| Vessel dilated | | | | 0.23 | | | | | | |
| Left anterior descending | 303 (35.3%) | 157 (39.5%) | 0.0874 | | 147 (40.4%) | 145 (39.8%) | 0.0114 | 317 (36.8%) | 140 (35.4%) | 0.0286 |
| Diagonal/Intermediate | 10 (1.2%) | 9 (2.3%) | 0.0848 | | 7 (1.9%) | 5 (1.4%) | 0.0423 | 14 (1.6%) | 6 (1.5%) | 0.0047 |
| Left circumflex | 100 (11.7%) | 37 (9.3%) | 0.0762 | | 36 (9.9%) | 35 (9.6%) | 0.0090 | 94 (10.9%) | 42 (10.7%) | 0.0077 |
| Right coronary | 353 (41.1%) | 146 (36.8%) | 0.0895 | | 135 (37.4%) | 138 (37.9%) | 0.0169 | 340 (39.5%) | 161 (40.9%) | 0.0303 |
| Left main/Multivessel | 88 (10.3%) | 45 (11.3%) | 0.0347 | | 37 (10.2%) | 39 (10.7%) | 0.0177 | 92 (10.7%) | 43 (11.0%) | 0.0074 |
| Bypass graft | 4 (0.5%) | 3 (0.8%) | 0.0371 | | 2 (0.5%) | 2 (0.5%) | 0.0000 | 4 (0.5%) | 2 (0.4%) | 0.0035 |
| Number of diseased vessels | | | | 0.94 | | | | | | |
| 1 | 361 (42.1%) | 166 (41.8%) | 0.0053 | | 150 (41.2%) | 152 (41.8%) | 0.0112 | 362 (42.1%) | 176 (44.6%) | 0.0512 |
| 2 | 232 (27.0%) | 108 (27.2%) | 0.0037 | | 103 (28.3%) | 97 (26.6%) | 0.0369 | 231 (26.8%) | 99 (25.2%) | 0.0374 |
| 3/Left main | 265 (30.9%) | 123 (31.0%) | 0.0021 | | 111 (30.5%) | 115 (31.6%) | 0.0238 | 268 (31.1%) | 119 (30.2%) | 0.0194 |

(*Continued*)

**Table 1.** (Continued)

| Variable | Original Sample | | | | Matched Sample | | | Weighted Sample | | |
|---|---|---|---|---|---|---|---|---|---|---|
| | No Morphine (n = 858) | Morphine (n = 397) | Absolute Standardized Difference | p Value | No Morphine (n = 364) | Morphine (n = 364) | Absolute Standardized Difference | No Morphine (n = 860) | Morphine (n = 394) | Absolute Standardized Difference |
| Thrombus aspiration | 331 (38.6%) | 204 (51.4%) | 0.2594 | <0.0001 | 163 (44.8%) | 179 (49.2%) | 0.0890 | 363 (42.2%) | 158 (40.1%) | 0.0430 |
| Initial TIMI flow | | | | 0.02 | | | | | | |
| 0 | 514 (59.9%) | 254 (64.0%) | 0.0840 | | 230 (63.2%) | 232 (63.7%) | 0.0114 | 529 (61.5%) | 249 (63.2%) | 0.0361 |
| 1 | 147 (17.1%) | 76 (19.1%) | 0.0522 | | 73 (20.1%) | 68 (18.7%) | 0.0348 | 153 (17.8%) | 74 (18.7%) | 0.0234 |
| 2 | 109 (12.7%) | 42 (10.6%) | 0.0663 | | 33 (9.1%) | 39 (10.7%) | 0.0552 | 102 (11.8%) | 36 (9.1%) | 0.0874 |
| 3 | 88 (10.3%) | 25 (6.3%) | 0.1441 | | 28 (7.7%) | 25 (6.9%) | 0.0317 | 77 (8.9%) | 35 (9.0%) | 0.0007 |
| Final TIMI flow | | | | 0.96 | | | | | | |
| 0 | 8 (0.9%) | 4 (1.0%) | 0.0077 | | 4 (1.1%) | 3 (0.8%) | 0.0282 | 9 (1.1%) | 4 (1.0%) | 0.0098 |
| 1 | 6 (0.7%) | 3 (0.8%) | 0.0066 | | 2 (0.5%) | 3 (0.8%) | 0.0333 | 6 (0.7%) | 2 (0.6%) | 0.0081 |
| 2 | 43 (5.0%) | 19 (4.8%) | 0.0105 | | 15 (4.1%) | 17 (4.7%) | 0.0268 | 40 (4.6%) | 16 (3.9%) | 0.0349 |
| 3 | 801 (93.4%) | 371 (93.5%) | 0.0038 | | 343 (94.2%) | 341 (93.7%) | 0.0231 | 805 (93.6%) | 372 (94.5%) | 0.0368 |
| Failed PCI | 26 (3.0%) | 11 (2.8%) | 0.0155 | 0.86 | 12 (3.3%) | 10 (2.7%) | 0.0327 | 26 (3.1%) | 14 (3.6%) | 0.0307 |

both matched and weighted samples. Adequate balance on baseline covariates has been achieved in both matched and weighted sets since potentially prognostically important covariates have been balanced between the treated and control groups (Table 1, Fig 1). Importance of each of the baseline variables included in the propensity score model is shown in Fig 2. It is of note, that symptom-onset-to-door time as a non-linear parameter was by far the most important predictor of allocation of treatment with intravenous morphine (Figs 2 and 3), followed by the Killip class, current smoking status, prehospital heparin and clopidogrel application, and use of aspiration thrombectomy. Patients were more likely to be treated with morphine when presenting 2 hours after symptom onset (Fig 3), being in Killip class 2 or 3, being active smokers, having received heparin and clopidogrel as well, and when aspiration thrombectomy was also performed. Median amounts of morphine hydrochloride applied in the treatment arms were 4.0 mg (IQR: 2.0 to 7.0 mg), 4.0 mg (IQR: 2.0 to 7.0 mg), and 4.0 mg (IQR: 2.0 to 6.2 mg) in the original, matched, and weighted data sets, respectively.

## Primary end point

**Original sample, crude analysis.** Comparison of the Kaplan-Meier survival curves using the log-rank test revealed a statistically significant absolute all-cause mortality risk difference between the control and treated cohorts favoring treatment with morphine (p = 0.0229, Fig 4, left panel). Similarly, analysis of the relative effect size using a naïve, univariable Cox model, the hazard ratio (HR) was 0.79, 95% CI: 0.64 to 0.97, p = 0.0233, (Fig 5, upper panel).

**Estimation of the average treatment effect for the treated (ATT) using propensity score matching.** After adjusting for confounding with 1:1 propensity score matching, there was no absolute risk difference detectable between the Kaplan-Meier survival curves of the control and treated groups (p = 0.3046, log-rank test stratified on matched pairs, Fig 4, middle panel). Likewise, the relative change in the hazard of death was not statistically significant when analyzed by Cox regression (HR: 0.98, 95% CI: 0.76 to 1.26, p = 0.8574, Fig 5, middle panel).

**Assessing the average treatment effect (ATE) by inverse probability of treatment weighting with stabilized weights.** As to absolute mortality risk difference, the Kaplan-

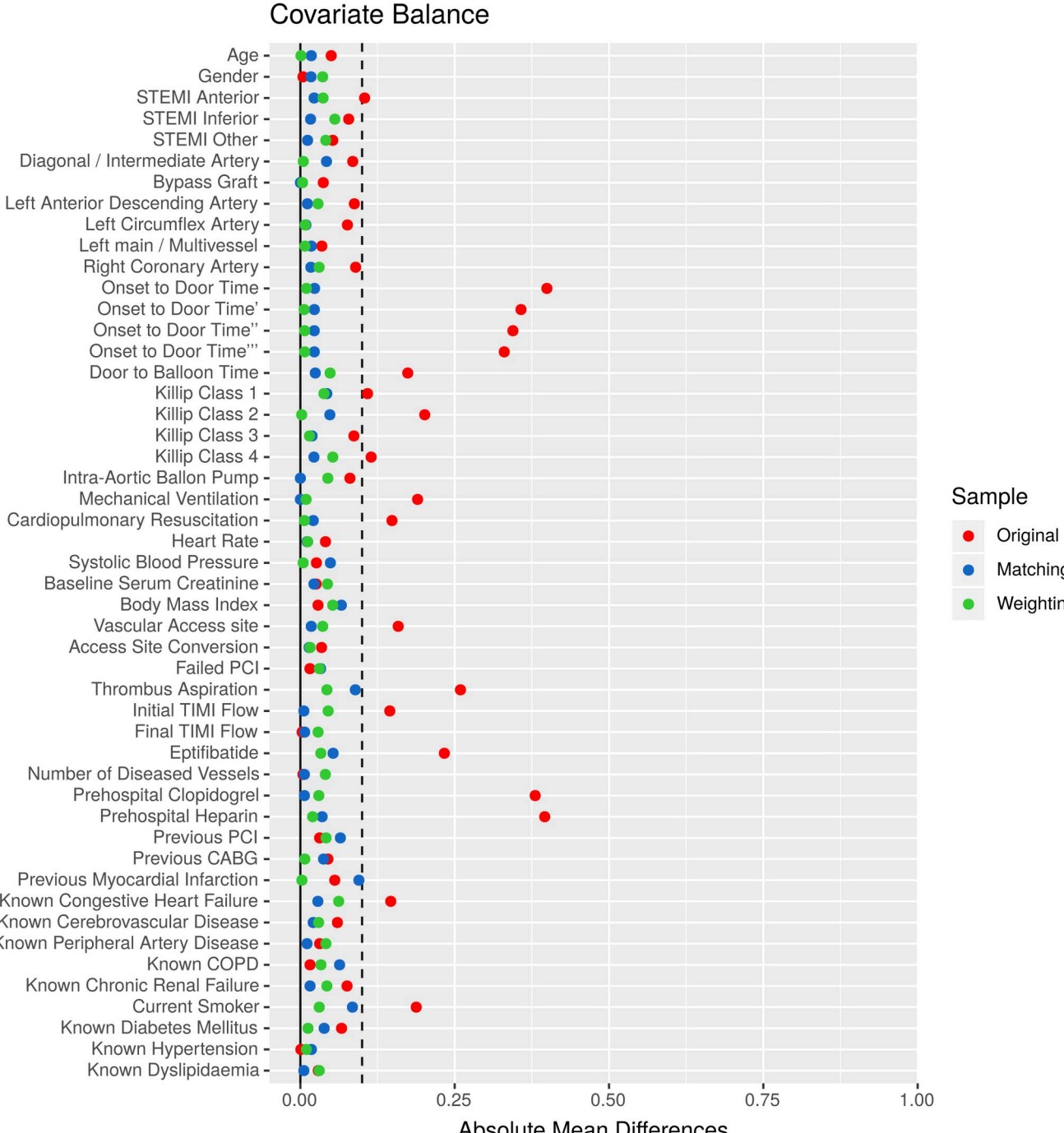

**Fig 1. Covariate balance.** The dot chart shows absolute standardized differences between control and treated groups across all measured baseline covariates. A value less than 0.1 was considered as an acceptable standardized bias. Systematic differences between treated and untreated patients in the original cohort have been eliminated in both matched and weighted samples. Adequate balance on baseline variables has been achieved in both matched and weighted sets since potentially prognostically important covariates have been balanced between the treated and control groups.

Meier curves of the treated and untreated arms were almost identical (p = 0.8518, design-based log-rank test, Fig 4, right panel). In addition, the hazard ratio was 1.01, 95% CI: 0.80 to 1.28, p = 0.9010 (Fig 5, lower panel).

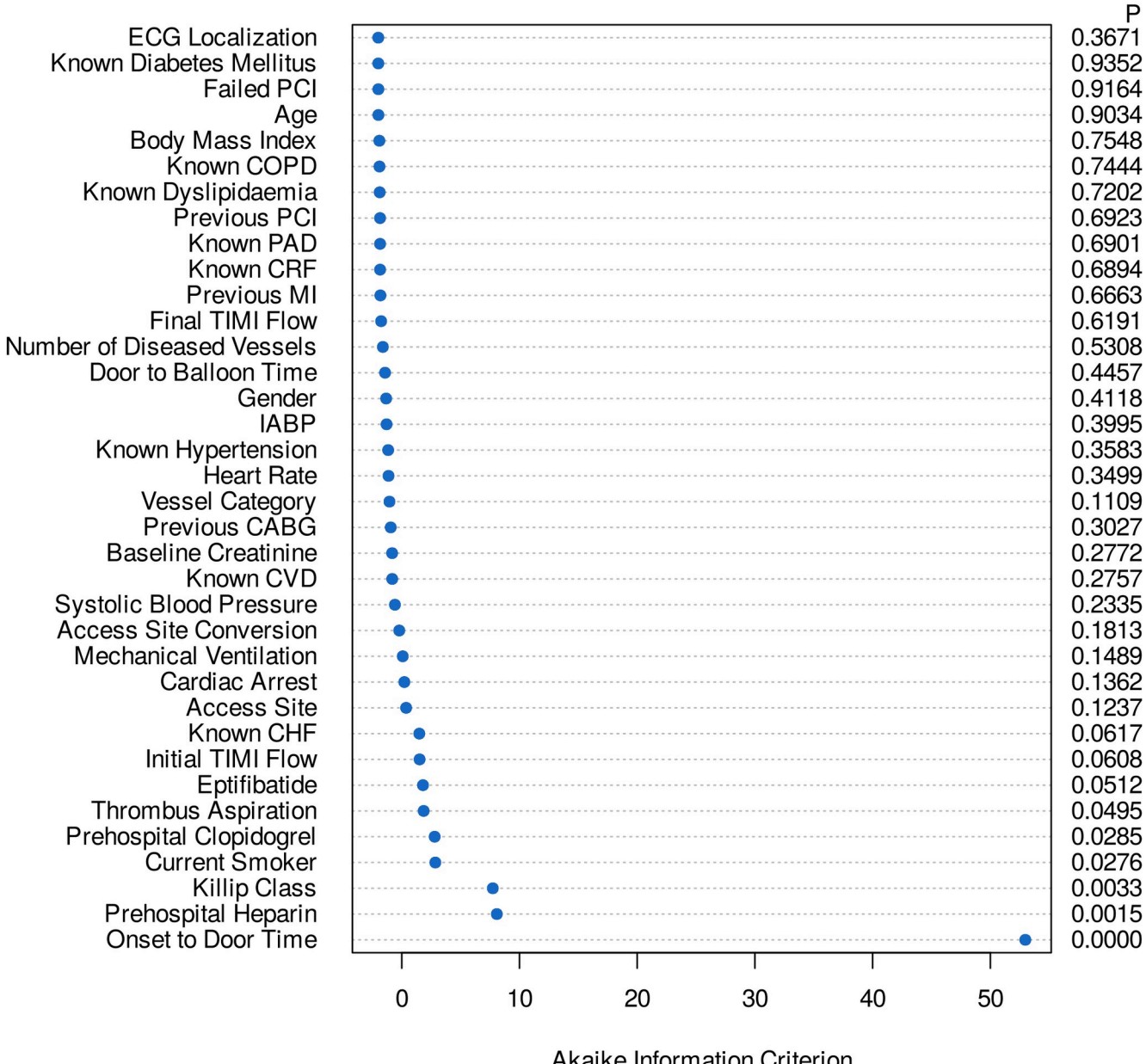

**Fig 2. Importance of variables in the propensity score models.** Dot chart depicts the importance of each variable as measured by the Akaike information criterion. The p value denotes statistical significance. The most important predictors of treatment allocation were onset-to-door time as a non-linear parameter, Killip class, current smoking status, prehospital heparin and clopidogrel application, and use aspiration thrombectomy.

## Secondary outcome measure

There was no difference in predischarge left ventricular ejection fraction between the control and treated groups–in both statistical and clinical senses–in any of the analyzed samples. The results are summarized in Table 2.

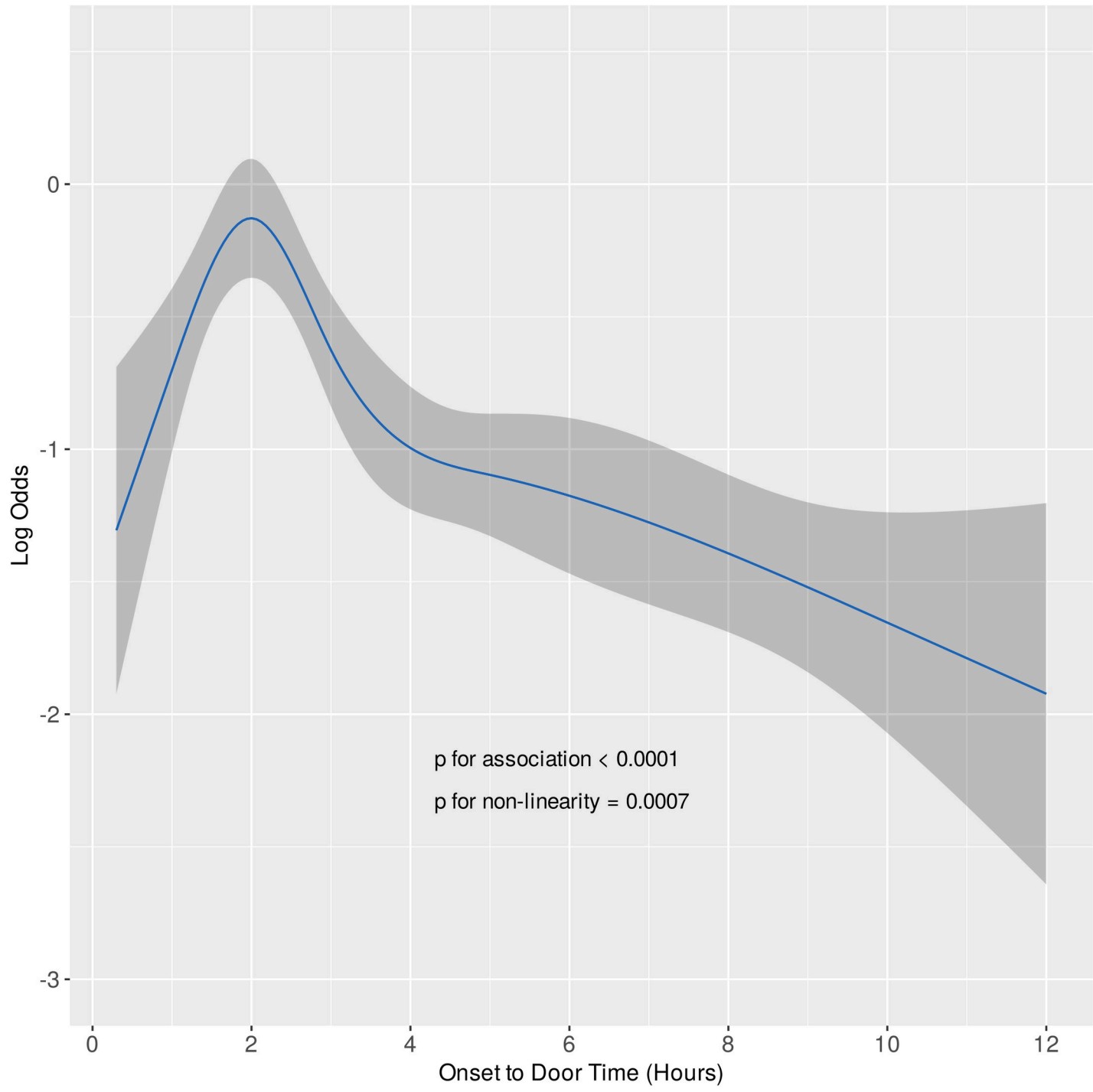

**Fig 3. Unadjusted association of onset-to-door time with intravenous morphine use.** The relationship was explored using a restricted cubic spline with five knots placed at 1, 2, 3, 5, and 10 hours (corresponding to percentiles 5, 27.5, 50, 72.5, and 95). With these settings, the curve is allowed to be flexible between 1 and 10 hours, representing 90% of the sample. The gray ribbon shows 95% confidence intervals. The association is highly significant (p<0.0001). Wald testing for linearity suggests a strong non-linear relationship (p = 0.0007).

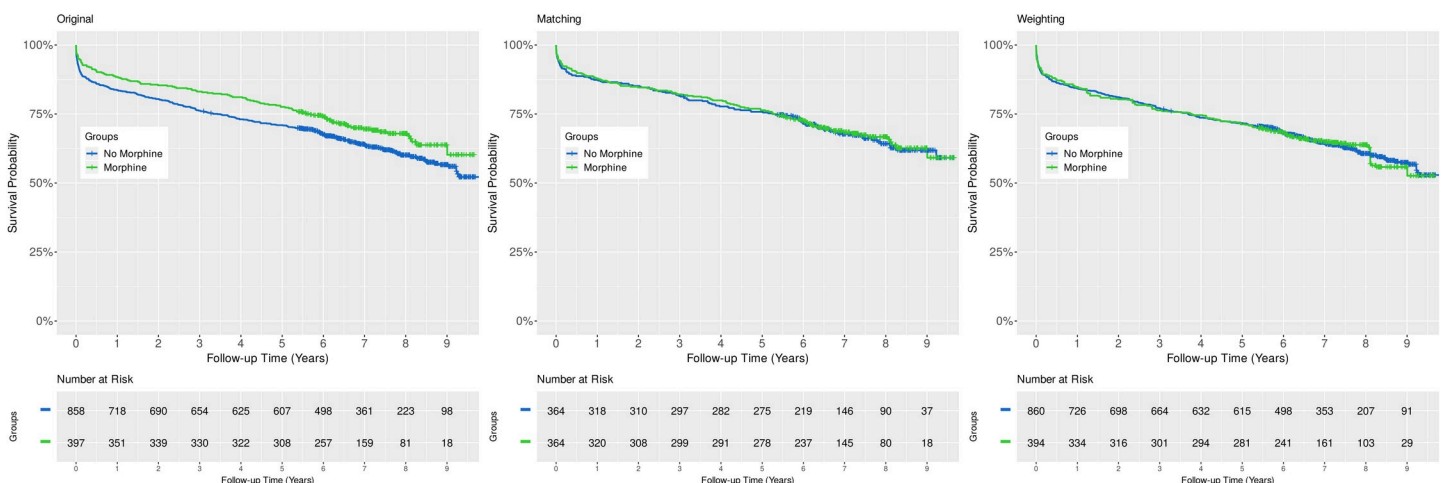

**Fig 4. Comparison of Kaplan-Meier survival curves.** Analysis of the crude data revealed a statistically significant absolute mortality risk difference between the control and treated groups (left panel, p = 0.0229, log-rank test). However, this difference is not detectable after adjusting for confounding using any of the applied methods (middle panel, propensity score matching, p = 0.3046, log-rank test stratified on matched pairs; right panel, inverse probability of treatment weighting, p = 0.8518, design-based log-rank test). Censored data are indicated with small vertical tick-marks.

## Discussion

### Principal findings, general considerations

Morphine is traditionally used in STEMI patients to relieve pain, decrease pulmonary congestion, and anxiety. However, according to in vitro measurements, intravenous morphine delays and diminishes the effects of all currently used oral platelet $P2Y_{12}$ receptor antagonists (i.e., clopidogrel, prasugrel, and ticagrelor) [3–9]. Consequently, the European Society of Cardiology published a warning note in its current guidelines on STEMI that this phenomenon may lead to early treatment failure [1]. Nevertheless, there are limited data available about the impact of this interaction on clinical outcomes. Therefore, we investigated the effect of peri-procedural morphine application on all-cause mortality in real-world STEMI patients who underwent primary PCI. We intentionally choose all-cause rather than cardiovascular mortality as an objective, unbiased primary end point [25]. Also, though periprocedural use of MO is single time-point intervention, we deliberately investigated long-term rather than short-term mortality, since initial observational reports suggested that application of MO may be associated with poorer myocardial reperfusion [10] and larger infarct size [11] whose deleterious effects on mortality may better be detected later. To adjust for confounding, two distinct propensity score-based procedures were performed to assess both average treatment effect (ATE) and average treatment effect for the treated (ATT). Among the most important predictors of treatment allocation were symptom-onset-to-door time and Killip class suggesting that the application of morphine was not based simply on default preferences of the treating physicians but rather driven by the actual clinical presentation of the patient. Our results indicate that intravenous morphine may have no impact on both absolute and relative measures of mortality in patients treated with primary PCI.

### Context with previous reports

The importance of the interaction between IV morphine administration and oral platelet $P2Y_{12}$ receptor inhibitors on clinical outcomes is poorly elucidated. There was only one randomized controlled trial conducted in this field, the "Influence of Morphine on

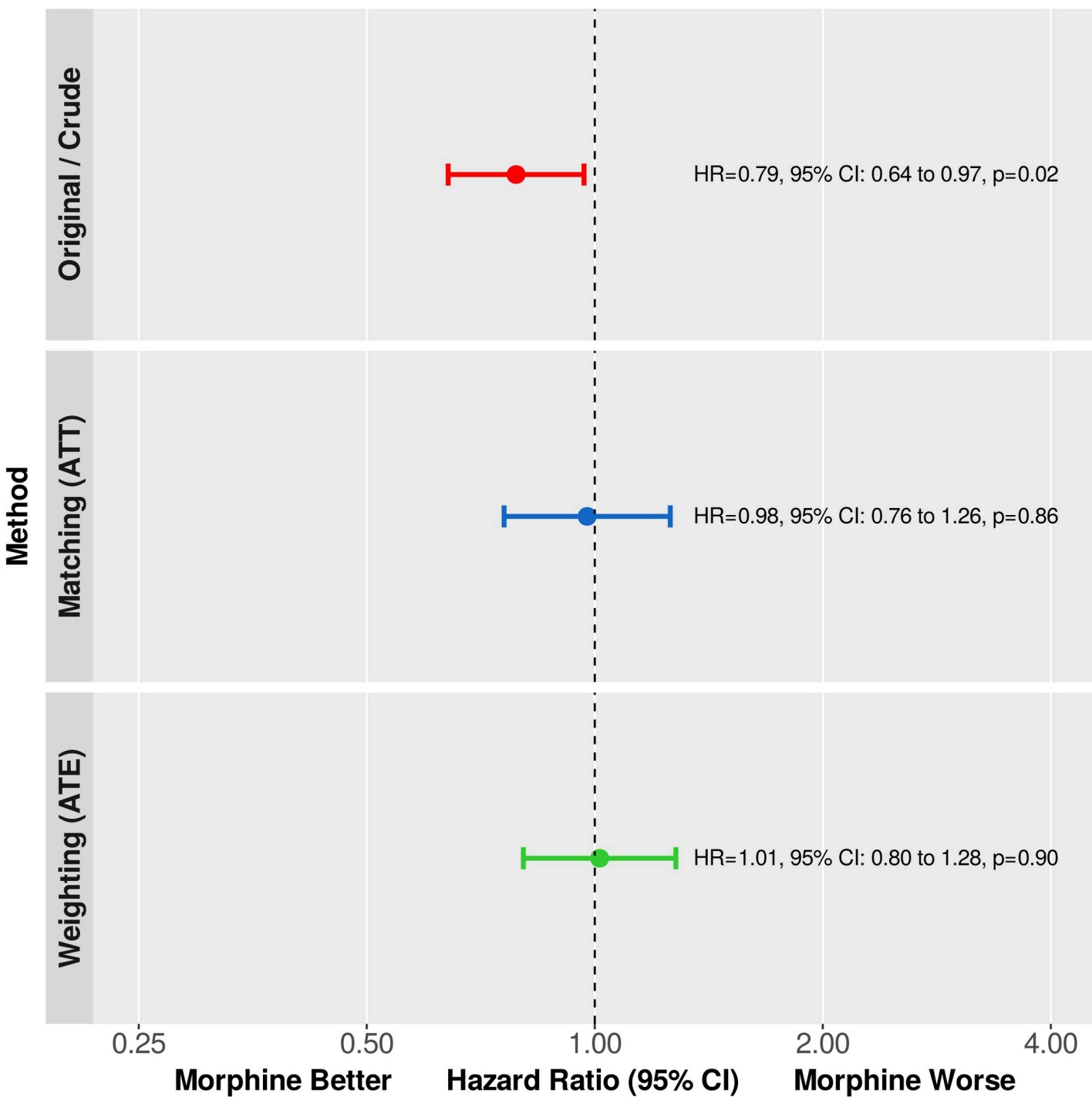

**Fig 5. Primary end point.** The relative change in the hazard of death was estimated using univariable Cox regression in the original, matched, and weighted samples. Hazard ratios (HR) are shown as point estimates and 95% confidence intervals. Analysis of the crude data showed a statistically significant relative mortality difference favoring treatment with morphine. However, after reducing the bias with propensity score matching or inverse probability of treatment weighting, there is no significant difference detectable—in both statistical and clinical senses.

Pharmacokinetics and Pharmacodynamics of Ticagrelor in Patients with Acute Myocardial Infarction" (IMPRESSION) study [6]. Beyond the in vitro finding that morphine delays and

**Table 2. Secondary outcome measure: Predischarge left ventricular ejection fraction.**

| Variable | Original Sample | | | Matched Sample | | | Weighted Sample | | |
|---|---|---|---|---|---|---|---|---|---|
| | No Morphine (n = 858) | Morphine (n = 397) | p Value | No Morphine (n = 364) | Morphine (n = 364) | p Value | No Morphine (n = 860) | Morphine (n = 394) | p Value |
| Predischarge Left Ventricular Ejection Fraction Median (IQR) (%) | 50.0 (43.0–56.0) | 50.0 (42.0–55.0) | 0.4580 | 50.0 (43.0–55.25) | 50.0 (42.0–55.0) | 0.7621 | 50.0 (42.5–55.0) | 50.0 (41.0–55.0) | 0.8612 |

attenuates ticagrelor exposure and action in patients with myocardial infarction (both STEMI and non-STEMI), the low number of in-hospital clinical events did not allow statistical analysis whereas longer-term outcomes were not recorded at all.

All other available data are observational (two post-hoc analyses of randomized controlled trials [17,18] and eight cohort studies [4,10–14,16,19] with mainly small to moderate sample sizes). Iakobishvili et al. published data of 249 propensity score-matched pairs showing that IV morphine use was associated with improved 30-day survival of STEMI patients (2.4% vs. 6.2%, p = 0.04 in the MO and no MO groups, respectively) [12]. In 2015, de Waha et al. reported data of 276 patients that IV morphine use is related to larger infarct size, greater extent of microvascular obstruction, and lower myocardial salvage index as found by cardiac magnetic resonance imaging (CMR). Yet, similarly to our results (Table 2), these differences could not be observed at the level of left ventricular ejection fraction. Also, in concert with our findings, survival curves were not different during the median follow-up of 16 months [11]. In the publication of Parodi et al. the small sample size (300 cases) did not allow to evaluate a potential detrimental consequence of IV morphine on in-hospital clinical end points. Yet, the published data do not imply such an effect [4]. According to the data of Puymirat et al. from 388 propensity score-matched pairs, prehospital morphine use in STEMI was not associated with worse in-hospital complications and 1-year mortality [13]. Likewise, in the small study by Bellandi et al. (182 cases) no change in complications could be observed during the hospital course that could be related to treatment with IV morphine [10]. Similarly, in the study by Gwag et al. with a sample size of 299 patients, there was no significant difference detectable in the clinical end point (a composite of cardiac death, recurrent myocardial infarction, ischemic stroke, and repeated coronary revascularization) according to IV morphine use with or without propensity score-matched analysis [16]. In addition, McCarthy et al. presented their results from a single-center observational study indicating that, after propensity score matching (107 pairs), morphine use do not affect in-hospital outcomes in STEMI patients [14]. Bonin et al. used the database of the "Does Cyclosporine Improve Outcome in ST Elevation Myocardial Infarction Patients" (CIRCUS) trial with 969 anterior STEMI patients [17,26]. They found no differences in a series of clinical end points including all-cause mortality rate during 1 year of follow-up [17]. Similarly, Lapostolle et al. performed a spin-off analysis of the "Administration of Ticagrelor in the Cath Lab or in the Ambulance for New ST Elevation Myocardial Infarction to Open the Coronary Artery" (ATLANTIC) study data [18,27]. There was no evidence that IV morphine application had an influence on any of the investigated clinical end points (all-cause death, myocardial infarction, stroke, urgent revascularization and definitive acute stent thrombosis) [18]. Also, Farag et al. did not detect any statistically significant changes in clinical event rates including death during hospital stay in their 2018 report with 300 patients [19].

More recently, Batchelor et al. published a meta-analysis of the above studies indicating that periprocedural intravenous morphine administration is not associated with adverse short-term clinical outcomes (in-hospital or 30-day myocardial reinfarction/mortality) in patients who undergo primary PCI [15]. Nevertheless, as described above, of the 11 investigated studies

10 were observational with predominantly small to moderate sample sizes and considerable methodological heterogeneity, whereas the remaining randomized controlled trial, because of the low sample size and short follow-up, was lacking any mortality events to be analyzed [6] making the interpretation of this meta-analysis equivocal. Also, the limited amount of data that are available about long-term outcomes were not sufficient for performing a meta-analysis.

In summary, our findings are consistent with all of the above reports suggesting that morphine administration does not increase the mortality in STEMI patients treated with primary PCI. Also, similarly to the results of de Waha et al. assessing the left ventricular ejection fraction with CMR [11], we could not detect any deterioration of the LVEF using echocardiography that could be attributable to IV morphine use.

## Strengths and limitations

To our knowledge, among the published papers investigating the impact of the interaction between intravenous morphine application and oral platelet P2Y$_{12}$ receptor inhibitors on all-cause mortality in patients treated with primary PCI, this study has the longest follow-up time (median 7.5 years, IQR: 6.5 to 8.6 years) and the highest number of events (457 deaths). Despite the observational nature of the present work, the long follow-up of a real-world population with an adequate number of events together with the applied complex statistical methods may allow an unbiased estimation of the treatment effect [24].

Our results are based on a prospective registry of a single institution. Also, we exclusively used a clopidogrel throughout the study period. Therefore, our findings may not be generalizable to populations/centers of other geographic regions and to other P2Y$_{12}$ receptor inhibitors. Lacking comprehensive long-term data on non-fatal ischemic events, we could not assess a possible effect of periprocedural intravenous morphine on them. Finally, we did not study (and discuss) data of non ST-segment elevation acute coronary syndrome (NSTE-ACS) cases because the inherent differences in the time frames of morphine/P2Y$_{12}$ inhibitor administration and the invasive procedure might have introduced substantial bias into the results. Thus, our data are not applicable for the setting of NSTE-ACS.

## Conclusions

Despite previous findings indicating that periprocedural intravenous morphine administration may delay and reduce the effect of oral platelet P2Y$_{12}$ receptor inhibitors in vitro which may be associated with larger infarct size, our data suggest that intravenous morphine may have no impact on predischarge left ventricular ejection fraction and–more importantly–on all-cause mortality in STEMI patients treated with primary PCI. Thus, it may safely be used for pain relief, pulmonary congestion, and anxiety even in the era of primary percutaneous coronary intervention, when reliable platelet P2Y$_{12}$ receptor inhibition is of crucial importance.

## Supporting information

**S1 Appendix. Extended methods: Statistical analysis.**
(PDF)

**S1 Dataset. Dataset.**
(CSV)

## Author Contributions

**Conceptualization:** Dominika Domokos, Istvan Hizoh.

**Data curation:** Dominika Domokos, Istvan Hizoh.

**Formal analysis:** Istvan Hizoh.

**Investigation:** Dominika Domokos, Istvan Hizoh.

**Methodology:** Istvan Hizoh.

**Project administration:** Dominika Domokos.

**Supervision:** Bela Merkely, Istvan Hizoh.

**Visualization:** Istvan Hizoh.

**Writing – original draft:** Dominika Domokos, Istvan Hizoh.

**Writing – review & editing:** Dominika Domokos, Andras Szabo, Gyongyver Banhegyi, Laszlo Major, Robert Gabor Kiss, David Becker, Istvan Ferenc Edes, Zoltan Ruzsa, Bela Merkely, Istvan Hizoh.

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
