## [Decision Letter · Decision Letter 0]

18 Dec 2020

PONE-D-20-27765

Impact of periprocedural morphine use on mortality in STEMI patients treated with primary PCI

PLOS ONE

Dear Dr. Hizoh,

Thank you for submitting your manuscript to PLOS ONE. After careful consideration, we feel that it has merit but does not fully meet PLOS ONE’s publication criteria as it currently stands. Therefore, we invite you to submit a revised version of the manuscript that addresses the points raised during the review process.

All issues raised by expert reviewers are required.

We look forward to receiving your revised manuscript.

Kind regards,

Vincenzo Lionetti, M.D., PhD

Academic Editor

PLOS ONE

Journal Requirements:

Reviewers' comments:

Reviewer's Responses to Questions

**Comments to the Author**

1. Is the manuscript technically sound, and do the data support the conclusions?

Reviewer #1: Yes

Reviewer #2: Yes

2. Has the statistical analysis been performed appropriately and rigorously? 

Reviewer #1: No

Reviewer #2: I Don't Know

3. Have the authors made all data underlying the findings in their manuscript fully available?

Reviewer #1: Yes

Reviewer #2: Yes

4. Is the manuscript presented in an intelligible fashion and written in standard English?

Reviewer #1: Yes

Reviewer #2: Yes

5. Review Comments to the Author

Reviewer #1: Interesting manuscript that analyses the effects of intravenous morphine on patients suffering of STEMI.

The follow up reported is important and gives high value to the manuscript, which is well written and with a sound methodology.

I have no particular observations. Just the suggestion to underline in the abstract (or in title ?) the fact that only copidogrel was in use in your center.

Reviewer #2: The paper is of sure interest for the readership of PLOS ONE, the topic is appealing and the structure is readable with ease. However, the narrative rationale sounds somewhat unbalanced. In particular, it juxtaposes extremely verbose paragraphs to utterly synthetic ones. In other words my suggestion is to encourage authors in improving the flow so to maintain a coherent register. Nevertheless, i believe the paper publishable pending minor revision.

My main point against the acceptance in its present form, is based upon the fact that the statistical part (which was frankly over my head) is given in such a synthetic way that is understandable only to a specialized readership. While, i believe important to make it accessible also to non expert and casual readers. In this regard, it is not clear why a test was chosen against an another one, or why the results of a given test are functional to the mastering of decision and assumption. In this way, the ground onto which the discussion section is built sounds rather speculative.

6. PLOS authors have the option to publish the peer review history of their article (what does this mean?). If published, this will include your full peer review and any attached files.

Reviewer #1: **Yes: **Prof. Edoardo De Robertis

Reviewer #2: No

---

## [Author Response · Author response to Decision Letter 0]

27 Dec 2020

Response to Reviewers

 We thank the Editor and the Reviewers for their constructive criticism. The insightful comments have been carefully addressed.

Journal Requirements:

1. Please ensure that your manuscript meets PLOS ONE's style requirements, including those for file naming. The PLOS ONE style templates can be found at https://journals.plos.org/plosone/s/file?id=wjVg/PLOSOne_formatting_sample_main_body.pdf

 and https://journals.plos.org/plosone/s/file?id=ba62/PLOSOne_formatting_sample_title_authors_affiliations.pdf

In the revised manuscript, we tried to accomplish the necessary changes according to PLOS ONE's style requirements.

Reviewers' comments:

Reviewer's Responses to Questions

Comments to the Author

1. Is the manuscript technically sound, and do the data support the conclusions?

Reviewer #1: Yes

Reviewer #2: Yes

2. Has the statistical analysis been performed appropriately and rigorously?

Reviewer #1: No

Reviewer #2:I Don't Know

Indeed, the applied statistical methods are complex and their detailed description may deteriorate the readability of the paper. Yet, this methodology is necessary to account for the observational nature of the data and may allow an unbiased estimation of the treatment effect. Therefore, we shortened the Statistical Analysis section for improved readability. Nevertheless, for the above reasons and to be in concert with the requirement that “Manuscripts submitted to PLOS ONE are expected to report statistical methods in sufficient detail for others to replicate the analysis performed” we now provide S1 Appendix with detailed methodology. Please see response to Reviewer #2’s comment bellow as well.

3. Have the authors made all data underlying the findings in their manuscript fully available?

Reviewer #1: Yes

Reviewer #2: Yes

In the revised manuscript, the analyzed data are available as Supporting Information (S2 Dataset).

4. Is the manuscript presented in an intelligible fashion and written in standard English?

Reviewer #1: Yes

Reviewer #2: Yes

5. Review Comments to the Author

Reviewer #1: Interesting manuscript that analyses the effects of intravenous morphine on patients suffering of STEMI. The follow up reported is important and gives high value to the manuscript, which is well written and with a sound methodology. I have no particular observations. Just the suggestion to underline in the abstract (or in title ?) the fact that only clopidogrel was in use in your center.

Thank you for the comment. The exclusive use of clopidogrel is now emphasized in the abstract as well. (It was also stated in the Methods and Limitation sections of the original manuscript.)

Reviewer #2: The paper is of sure interest for the readership of PLOS ONE, the topic is appealing and the structure is readable with ease. However, the narrative rationale sounds somewhat unbalanced. In particular, it juxtaposes extremely verbose paragraphs to utterly synthetic ones. In other words my suggestion is to encourage authors in improving the flow so to maintain a coherent register. Nevertheless, i believe the paper publishable pending minor revision.

My main point against the acceptance in its present form, is based upon the fact that the statistical part (which was frankly over my head) is given in such a synthetic way that is understandable only to a specialized readership. While, i believe important to make it accessible also to non expert and casual readers. In this regard, it is not clear why a test was chosen against an another one, or why the results of a given test are functional to the mastering of decision and assumption. In this way, the ground onto which the discussion section is built sounds rather speculative.

Thank you for this important remark. Indeed, the applied statistical methods are complex and their detailed description may worsen the readability. Yet, we deliberately sought to use modern, complex statistical methods that may allow an unbiased assessment of the treatment effect despite the observational nature of the data. In the revised manuscript, we shortened the Statistical Analysis section for better legibility. Nevertheless, for the above reasons and to be in concert with the requirement that “Manuscripts submitted to PLOS ONE are expected to report statistical methods in sufficient detail for others to replicate the analysis performed” we now present S1 Appendix with detailed methodology. We think that further explanation of the chosen methodology would be beyond the scope of the present paper. Nevertheless, we include references with DOI numbers / hyperlinks for the reader wishing greater insight into the applied statistical methods. If any further doubt remains regarding the appropriateness of the statistical analysis, we suggest performing a formal statistical review of the manuscript.

---

## [Decision Letter · Decision Letter 1]

4 Jan 2021

Impact of periprocedural morphine use on mortality in STEMI patients treated with primary PCI

PONE-D-20-27765R1

Dear Dr. Hizoh,

We’re pleased to inform you that your manuscript has been judged scientifically suitable for publication and will be formally accepted for publication once it meets all outstanding technical requirements.

Kind regards,

Vincenzo Lionetti, M.D., PhD

Academic Editor

PLOS ONE

Additional Editor Comments (optional):

Reviewers' comments:

Reviewer's Responses to Questions

**Comments to the Author**

1. If the authors have adequately addressed your comments raised in a previous round of review and you feel that this manuscript is now acceptable for publication, you may indicate that here to bypass the “Comments to the Author” section, enter your conflict of interest statement in the “Confidential to Editor” section, and submit your "Accept" recommendation.

Reviewer #2: All comments have been addressed

2. Is the manuscript technically sound, and do the data support the conclusions?

Reviewer #2: Yes

3. Has the statistical analysis been performed appropriately and rigorously? 

Reviewer #2: I Don't Know

4. Have the authors made all data underlying the findings in their manuscript fully available?

Reviewer #2: Yes

5. Is the manuscript presented in an intelligible fashion and written in standard English?

Reviewer #2: Yes

6. Review Comments to the Author

Reviewer #2: (No Response)

7. PLOS authors have the option to publish the peer review history of their article (what does this mean?). If published, this will include your full peer review and any attached files.

Reviewer #2: No

---

## [Editor Report · Acceptance letter]

5 Jan 2021

PONE-D-20-27765R1 

Impact of periprocedural morphine use on mortality in STEMI patients treated with primary PCI 

Dear Dr. Hizoh:

I'm pleased to inform you that your manuscript has been deemed suitable for publication in PLOS ONE. Congratulations! Your manuscript is now with our production department. 

Kind regards, 

on behalf of

Prof. Vincenzo Lionetti 

Academic Editor

PLOS ONE